# Validity of the "Big Data Tendency in Education" Scale as a Tool Helping to Reach Inclusive Social Development

**Antonio Matas-Terrón** [1] , **Juan José Leiva-Olivencia** [2,*], **Pablo Daniel Franco-Caballero** [1] **and Francisco José García-Aguilera** [3]

1   Department of Methods of Researching in Education, University of Málaga, 29071 Málaga, Spain;
    amatas@uma.es (A.M.-T.); pablo.franco@uma.es (P.D.F.-C.)
2   Department of Didactic and Organization in School, University of Málaga, 29071 Málaga, Spain
3   Department of Theory and History of Education, University of Málaga, 29071 Málaga, Spain;
    fjgarciaa@uma.es
*   Correspondence: juanleiva@uma.es

**Abstract:** Big Data technology can be a great resource for achieving the Sustainable Development Goals in a fair and inclusive manner; however, only recently have we begun to analyse its impact on education. This research goal was to analyse the psychometric characteristics of a scale to assess opinions that educators in training have about Big Data besides their related emotions. This is important, as it will be the educators of the future who will have to manage with Big Data at school. A nonprobability sample of 337 education students from Peru and Spain was counted. Internal consistency, as well as validity, were analysed through exploratory and confirmatory factorial analysis. The results show good psychometric values, highlighting as relevant a latent structure of six factors that includes emotional and cognitive dimensions. As a result, the profile defining the participants in relation to Big Data was identified. Finally, the implications of the Big Data for Inclusive Education in a sustainable society are discussed.

**Keywords:** big data; education students; Peru; Spain; confirmatory factor analysis; insight scale

## 1. Introduction

Big Data has become an emerging term and concept from a social, cultural and pedagogical point of view. Although the term has been used since the 1980s, it was in 2008 that D.J. Partir (from LinkedIn) and Jeft Hammerbadier (from Facebook) used "Big Data" to refer to a new professional activity. The paper "The Exabyte era" in "Wired" magazine in 2010, as well as similar papers in that year, determined the beginning of Big Data as social and business phenomena [1]. Now, Big Data admits a large number of definitions and perspectives [2] mainly related to their technological properties. In this sense, Big Data would be defined as managing, gathering and organizing big volumes of data, and how they are analysed and interpreted [3]. The value generated by the data can be considered the most important element of the characteristics of Big Data [4].

Big Data could be the key to a new social revolution [5]. Nevertheless, much of the literature is focused on the business world [6] rather than areas such as education. This is quite remarkable, considering that the implementation of Big Data in the educational system could be a real boost in terms of inclusion and improving the quality of teaching and learning processes [7]. This feature is affecting to the core of the Inclusive School or Inclusive Education. An inclusive school is one where all students feel included and accepted, whatever their abilities, recognized in their uniqueness, valued and with the possibility of participating in the school. In fact, the integration of Big Data in Education

is turning into a cornerstone for the Inclusive School, as the Khan Academy has been showing for the last 15 years (see khanacademy.org). In this sense, it has been mentioned that the benefits of Big Data technology in Education would be linked to the improvement of education by helping personalized training, guiding students towards the programs that best fit their needs, effectively linking students to the labour market, making educational funding more transparent, and improving the administration of the educational system [8]. Big Data can help improve the quality and well-being of students, especially those who are most vulnerable or who have learning issues.

Considering this situation, the idea behind the text is that Big Data will be important in education, its impact will be positive if educators manage it properly, and it will be managed properly if educators receive well-supported training. Then, the first step is to have an assessment instrument for evaluating educators in training. Improving the knowledge from the base through Big Data, the chances to achieve social sustainability and an inclusive school will be higher. Besides, this instrument must measure the level of knowledge, opinions and emotions that educators have in relation to this technology, to be sure that Big Data will be used correctly as some authors advise [2,4,5]. In this sense, some basic questions arose: How much do students of education know about Big Data? How willing are they to take on such technology? In the literature consulted, there were no answers to those questions that were sufficiently supported by empirical studies. For this reason, it was proposed to put forward a researching line about Big Data technology in education.

Besides, this topic is even more important in the south of Spain, where some universities (e.g., University of Malaga) are debating about incorporating Big Data concepts in the syllabus of education students. On the other hand, the traditional collaboration between Andalusian (South of Spain region) and Latin-American Universities defines the satisfactory framework to build partnerships with South-American universities to study this issue.

This article's authors have the mission of developing a training activity in two Spanish universities and two other universities in Peru to improve the education students' skills about Big Data in order to get a more inclusive school in the future. For that aim, it is necessary to know not only the knowledge of the potential students regarding Big Data, but their opinions, attitudes, and emotions too [6], to design the training properly.

Then, the first stage was to find an instrument to measure the dimensions of interest (knowledge, attitude, emotions and opinions regarding Big Data in higher education students). A search in Scopus and Web of Science was conducted at the beginning of the research (February 2019) using this algorithm: (key (big and data) and key (high and education) and TITLE-ABS-KEY (questionnaire)). Only four documents were found and none had a useful instrument for our purpose. However, it is highlighted that there are other approaches, like the Project Analyzing Big Education Data (PABED)[7] project, that are worth taking into account since they try to incorporate the concept of Big Data in Education into practice. Nevertheless, this project does not have any instruments that could be adapted to our research. In short, the literature provided little documentation of instruments to gather information about Big Data in Education, although instruments related to the assimilation of new technologies, models and methodologies in education can be found [8–13].

Finally, in relation to Big Data in Education focusing on educators, students and managers, only the Assessment of "Big Data" Development in Education (VABIDAE in Spanish) questionnaire was found [14] after in-depth searching on the Internet. The VABIDAE survey is registered under a Creative Commons license and more information is available at https://vabidae.gitlab.io/vabidae/. Although this questionnaire has been used successfully in another study [15], this instrument has not been validated psychometrically. Then, as the first stage of our line of research, the highest priority was to validate this instrument. Therefore, the goal of this study was to examine several characteristics of the VABIDAE, including internal consistency and its factorial structure. In this sense, an exploratory factor analysis (EFA) was conducted, followed by a confirmatory factorial analysis (CFA). Another goal was to explore the profile of sample according to VABIDAE, taking into account that the sample can be from different cultures.

## 2. Methodology

Taking into consideration the arguments presented above, a pilot study was conducted. The target population was Spanish education students from the south of Spain and from north of Peru. Peru is a region of interest because of broad partnerships with the Andalusian universities. Nevertheless, an incidental sample of 337 participants were recruited from Peruvian and Spanish university students (mean age = 23.1, S.D. = 7.88, 70.03% female) of educational degrees. The majority of the participants are from Spain (70.03%) and 29.97% from Peru. The Spanish sample is from three universities: Malaga (107), Jaen (57) and Seville (72); while the Peruvian sample has students from six universities: Alas Peruanas (27), Pedro Ruiz Gallo (12), USAT (50) and Nacional de Piura (12). All Peruvian universities are from the same region (north Peru). Of the sample, 65.28% said that they had no idea about what Big Data is, 31.15% said that they knew a little bit but not enough, and only 3.25% said they knew what it is. All participants were informed of the research goals, giving their informed consent.

Big Data Applied to Education Scale (VABIDAE) [14] is a 31 items questionnaire that gathers information about opinions and how people face and perceive the presence of Big Data technologies in the educational system and in classrooms. The VABIDAE construction process is described by authors on the website https://vabidae.gitlab.io/vabidae/. As part of the VABIDAE, and previous to the questions, the scale embeds a short video about Big Data in Education. This video is available online by Euronews: https://es.euronews.com/2015/05/22/big-data-al-servicio-de-la-educacion.

This video contextualizes the topic and offers a piece of common information to all participants. This strategy is used to reduce misunderstandings and mistakes about what the issue is, as authors say. In this sense, the VABIDAE authors assume that those surveyed could have no previous idea or have misunderstanding ideas regarding what Big Data is, biasing their answer in order to keep the coherence between their wrong assumptions and their answers (more information about this topic at VABIDAE website). On the other hand, they highlight that this video was selected because it is from an official European mass media company, and because it is centred on the concept of Big Data in Education specifically.

After watching the video, participants started answering the items. The scale contains three subscales: (1) assessment of positive aspects of Big Data applied to education, (2) assessment of negative aspects and (3) emotions that Big Data induces in those surveyed. Participants rated their agreement with each item on a 5-point scale: positive and negative issues (1 Not at all, 2 I think not, 3 I don't know, 4 I think so, and 5 I strongly agree) and emotional items (1 Nothing at all, 2 Almost nothing, 3 I don't know/I'm indifferent, 4 Something, 5 Totally).

The instrument is completed with a series of sociodemographic questions (age, gender, residence, university, etc.) and the question "Are you inclined to use Big Data in your future job as an Educator?"

The procedure started with an agreement with Spanish and Peruvian university teachers contacted via email. They informed their students about this research and asked for their voluntary participation. The data was collected from March to November 2019 via an online Form Application by Google. The form includes instructions for respondents regarding how to distribute and answer it. For this purpose, the online form was available with a password given to professors and teachers of the participating students. After the period of collecting data, the application remained closed, preventing uncontrolled access.

An exploratory factor analysis (EFA) was conducted to examine the psychometric properties of VABIDAE. After checking that the sample was big enough (more than 150 cases and at least 5 cases for each variable), as is proposed by Pallant [16], the EFA was conducted. Specifically, the principal component axis factoring with Equamax rotation was conducted to examine whether subscales emerged and to analyse the items' consistency, in accordance with the advice by Carretero-Dios and Pérez [17]. Equamax was developed in order to maximize loads in both components and variable. On the other hand, the eigenvalue over one and the scree test were considered as criteria in order to extract the number of factors. In this sense, loads below 0.40 were considered as low [18,19].

Following, a confirmatory factorial analysis (CFA) was conducted based on the previous exploratory analysis result. A CFA provides a more powerful method than EFA to determine the best-fitting factor structure of the scale, because individual items are a priori, predicted to only load on their theoretically driven latent variables rather than loading on all latent variables in the exploratory factor analysis [20]. A maximum likelihood estimation (MLM) was conducted because it has been shown to perform well even under non-normal conditions [21]. This CFA was developed without splitting the sample into two groups, considering it more important to use all data than comparison between results from two analytical strategies. This approach was considered more coherent with the aim of this study, what it is to identify the empirical validity of VABIDAE. On the other hand, both EFA and CFA results were taken into consideration, being contrasted to each other as a triangulation methodological approach. This strategy facilitates the identification of overfitting in this case.

The following goodness-of-fit indexes were developed: ratio Chi-square-degree freedom, where ratios less than 3 are considered acceptable [22]; comparative fit index (CFI) obtained from a free-distribution estimation due to the ordinal scale of the observed variables, where values greater than 0.95 indicate a good fit of the model [23]; root-mean-square error of approximation (RMSEA), where values smaller than 0.05 indicate a good fit of the model and values up to 0.08 represent a reasonable error of approximation to the population [24]; and standardized root-mean-square residual (SRMR), where values smaller than 0.08 indicate a good fit of the model [25,26].

Next, the reliability of the subscales was conducted using Cronbach's alpha coefficient ($\alpha$) and McDonald's omega coefficient ($\Omega$). Then, all VABIDAE scores on the subscales were calculated by averaging their items.

The convergence and divergence validity were not calculated, because no other scale measuring Big Data attitude was found. Using any other instrument would involve making a decision without enough evidence to compare to VABIDAE. For the analysis, the R version 3.6.1 (R core Team, 2019) and SPSS version 24 were used.

## 3. Results

### 3.1. Exploratory Factor Analysis

An exploratory factor analysis was conducted using the principal component analysis approach. The Kaiser-Mayer-Olkin measure was 0.90, Bartlett's test of sphericity was statistically significant (Chi-square = 5672.291; d.f. = 465; $p < 0.001$) and the determinant of the correlation matrix was practically 0 (D = 2.61E-008). These results suggest that sampling adequacy was acceptable and that factor analysis is appropriate for the data.

The analysis offers six component solutions with 64.6% explained variance after extraction. The results after the Equamax rotation is available in Table 1. Every component was given a title and an interpretation depending on the items loading on them. To elaborate the meanings, every item was interpreted within the component where it had the highest charge.

1.  Negative feeling: negative emotions and emotional states that appear when thinking about Big Data, such as guilty, angry, shame, etc.
2.  Negative impacts: how Big Data could have negative social consequences, mainly related to the educational system and democracy.
3.  Positive impacts: benefits in the educational results because of Big Data.
4.  Educational system improvement: benefits in educational organization and teacher recruitment.
5.  Positive feelings: good emotions and emotional states related to Big Data in school.
6.  Privacy: concerning in relation to privacy loss and a possible increase in governmental control.

**Table 1.** The components were interpreted as the following.

| | | 1 Negative Feelings | 2 Negative Impacts | 3 Positive Impacts | 4 Educational System Improving | 5 Positive Feelings | 6 Privacy |
|---|---|---|---|---|---|---|---|
| v27 | I feel shame | 0.880 | - | - | - | - | - |
| v28 | I feel guilty | 0.836 | - | - | - | - | - |
| v25 | It makes me feel angry | 0.745 | - | - | - | - | - |
| v30 | I feel helpless | 0.709 | - | - | - | - | - |
| v26 | It makes me anxious | 0.664 | - | - | - | - | - |
| v31 | It makes me bored | 0.563 | - | - | - | - | - |
| v21 | Control of the education system by the company | - | 0.775 | - | - | - | - |
| v20 | Control of the education system by governments | - | 0.766 | - | - | - | - |
| v18 | Increase in power of politicians | - | 0.707 | - | - | - | - |
| v19 | System manipulation | - | 0.704 | - | - | - | - |
| v17 | Increase of power of centre managers | - | 0.568 | - | - | - | - |
| v15 | Computer attacks | - | 0.462 | - | - | - | - |
| v16 | Loss of teacher functions | - | 0.456 | - | - | - | - |
| v14 | Loss of the school's own socialization | - | 0.442 | - | - | - | - |
| v1 | Better meet the needs of students | - | - | 0.760 | - | - | - |
| v3 | Customize Education | - | - | 0.688 | - | - | - |
| v2 | Improve academic results | - | - | 0.654 | - | - | - |
| v10 | Promote educational quality in general | - | - | 0.622 | 0.455 | - | - |
| v8 | Produce educational resources adapted to students | - | - | 0.591 | - | - | - |
| v11 | Help prevent school failure | - | - | 0.543 | 0.425 | - | - |
| v7 | Improve teacher selection | - | - | - | 0.730 | - | - |
| v6 | Improve the organization of schools | - | - | 0.480 | 0.551 | - | - |
| v9 | Facilitate political decision making | - | - | - | 0.549 | - | - |
| v4 | Improve employability | - | - | - | 0.495 | - | - |
| v5 | Avoid plagiarism | - | - | - | 0.470 | - | - |
| v23 | It gives me hope | - | - | - | - | 0.774 | - |
| v24 | It makes me proud | - | - | - | - | 0.727 | - |
| v29 | It brings me relief | - | - | - | - | 0.589 | - |
| v22 | The subject amuses me | - | - | - | - | 0.587 | - |
| v13 | Loss of teacher privacy | - | - | - | - | - | 0.889 |
| v12 | Loss of student privacy | - | - | - | - | - | 0.728 |
| Explained variance | - | 23.511 | 19.406 | 10.023 | 4.311 | 4.001 | 3.347 |

*3.2. Confirmatory Factor Analysis*

From data, two models were tested. The first examined the six-factor fit from the Principal Component Analysis (PCA); the second tested the three-component structure according to the three original VABIDAE subscales.

Table 2 has the fit index from both models. The first six-component model provides a good fit to the data, with high RMSEA and SRMR indexes, although only the Chi-square suggests an acceptable fit. On the other hand, the Bentler CFI also suggests a fit, although low.

**Table 2.** The components were interpreted as the following.

| Models | Uncorrected Fit Statistics | | | | | |
|---|---|---|---|---|---|---|
| | Chi-Square (df; Pr) | RMSEA | Bentler CFI | SRMR | AIC | BIC |
| Six-factor model | 856.139 (419; <0.001) | 0.055 | 0.919 | 0.061 | 1010.139 | −1582.475 |
| Three-factor model | 1752.414 (431; <0.001) | 0.095 | 0.755 | 0.135 | 1882.414 | −756.042 |

Regarding the three-component model, only the RMSEA index suggests an acceptable fit, while the rest of the indexes show a poor fit. In this sense, the AIC, BIC and ratio Chi-square/degree freedom statistics support that the first model is better. Overall, these results say that the six-component model is significantly better than the second three-component model based on the theoretical structure and that fits the data well. The six-component correlations are shown in Table 3 and model parameters are available in the annex. Both models can be well interpreted from Psychopedagogist theories, such as a three-model attitude, the theory of reasoned action, etc. Nevertheless, the six-factor model is retained because of its better CFA indexes, and because it offers more specificity. The Parameter Estimates are available in Appendix A.

**Table 3.** The components were interpreted as the following.

| | Negative Feelings | Negative Impacts | Positive Impacts | Educational System Improving | Positive Feelings |
|---|---|---|---|---|---|
| Negative feelings | — | | | | |
| Negative impacts | 0.298 *** | — | | | |
| Positive impact | −0.125 * | 0.080 | — | | |
| Educational system improving | −0.010 | 0.108 * | 0.699 *** | — | |
| Positive feelings | −0.138 * | −0.132 * | 0.570 *** | 0.526 *** | — |
| Privacy | 0.174 ** | 0.569 *** | 0.005 | 0.006 | −0.167 * |

Note: * $p < 0.05$; ** $p = 0.01$; *** $p < 0.001$.

### 3.3. Item Characteristics and Internal Consistency

Because the six-component model had the best fit, both descriptive statistics and internal consistency were analysed, global scale and subscales. The internal reliability for the global VABIDAE scale and subscales was analysed using the Cronbach's alpha coefficient and McDonald's omega coefficient. The Cronbach's alpha was 0.86 and the McDonald's omega 0.873. Then, a good internal consistency from the VABIDAE measurement was considered. It was tested that the consistency does not improve dropping any item. The internal consistency was tested for every subscale too (see Table 4). No coefficients improved by dropping any item, so all of them were kept.

**Table 4.** The components were interpreted as the following.

| Subscale | Mean | McDonald's Omega | Cronbach's Alpha |
|---|---|---|---|
| Negative feelings | 2.025 | 0.883 | 0.878 |
| Negative impacts | 3.441 | 0.883 | 0.880 |
| Positive impacts | 3.441 | 0.883 | 0.880 |
| Educational improving | 3.623 | 0.818 | 0.815 |
| Positive feelings | 3.623 | 0.818 | 0.815 |
| Privacy | 3.320 | 0.824 | 0.824 |

### 3.4. Descriptive Statistics from Scale and Subscales

Finally, the descriptive statistics for subscales by country were calculated in order to know how the sample was according to VABIDAE. Results are available in Table 5.

**Table 5.** The components were interpreted as the following.

| | Pais | Negative Feelings | Negative Impacts | Positive Impacts | Educational Improvement | Positive Feelings | Privacy |
|---|---|---|---|---|---|---|---|
| Mean | Spain | 12.8 | 28.0 | 23.9 | 17.4 | 12.5 | 6.81 |
| | Peru | 10.6 | 26.4 | 25.9 | 19.9 | 15.1 | 6.24 |
| Std. error mean | Spain | 0.350 | 0.404 | 0.291 | 0.242 | 0.223 | 0.133 |
| | Peru | 0.553 | 0.712 | 0.459 | 0.423 | 0.373 | 0.234 |
| Standard deviation | Spain | 5.37 | 6.20 | 4.47 | 3.71 | 3.43 | 2.05 |
| | Peru | 5.56 | 7.15 | 4.61 | 4.25 | 3.75 | 2.35 |
| Minimum | Spain | 6.00 | 8.00 | 6.00 | 7.00 | 4.00 | 2.00 |
| | Peru | 6.00 | 8.00 | 7.00 | 5.00 | 5.00 | 2.00 |
| Maximum | Spain | 28.0 | 40.0 | 30.0 | 25.0 | 20.0 | 10.0 |
| | Peru | 30.0 | 40.0 | 30.0 | 25.0 | 20.0 | 10.0 |
| Skewness | Spain | 0.492 | −0.513 | −1.16 | −0.142 | −0.380 | −0.344 |
| | Peru | 1.72 | −0.388 | −2.27 | −1.10 | −0.828 | −0.122 |
| Std. error skewness | Spain | 0.158 | 0.158 | 0.158 | 0.158 | 0.158 | 0.158 |
| | Peru | 0.240 | 0.240 | 0.240 | 0.240 | 0.240 | 0.240 |
| Shapiro-Wilk p | Spain | <0.001 | <0.001 | <0.001 | 0.022 | < 0.001 | <0.001 |
| | Peru | <0.001 | 0.018 | <0.001 | < 0.001 | <0.001 | <0.001 |
| Student's t | | 3.44 *** | 2.13 ** | −3.71 *** | −5.38 *** | −6.30 *** | 2.26 (a) ** |
| Welch's t | | 3.39 *** | 2.01 ** | −3.66 *** | −5.09 *** | −6.08 *** | 2.14 ** |
| Cohen's d | | 0.409 | 0.253 | −0.441 | −0.639 | −0.749 | 0.269 |

(a) Levene's test is significant ($p < 0.05$); ** $p < 0.05$; *** $p < 0.001$.

Table 5 shows that subscales are biased, and that there are statistical differences between countries. In general, scores from Peru are more extreme as Table 4 shows. In this sense, the positive feelings average is significantly higher in Peru than in Spain; meanwhile, the average is less for negative feelings. In the same way, the educational improvement factor has a higher average in Peru than in Spain. On the other hand, the negative impacts factor has a similar average, although dispersion is higher in Peru, hence that Student's *t* is statistically significant. Eventually, Student's *t* test analysis was conducted using gender as a factor, although no statistical differences between male and female results at alpha 0.01 were found.

## 4. Discussion

The results show that VABIDAE has high internal consistency measurement, showing coherence about the Big Data effects, both positive (teaching improvement, curricular adaptation, etc.) and negative (privacy issues, isolation, etc.). Likewise, based on exploratory factor analysis, the internal structure of the instrument could be considered as valid. However, the original theoretical structure of the instrument needs to be revised, as the six-factor model is supported by confirmatory factor analysis.

The latent structure suggested by the EFA and the CFA recap the issues highlighted about Big Data by the reviewed literature, including negative and positive issues, both emotional dimension and in the impacts, opportunities and threats of Big Data in society, and specifically in education. [27]. However, the "improvement of education" and "privacy" emerge as new topics, suggesting that participants give special importance to these issues. It is important to point out that the "improvement of education" has a positive aspect and "privacy" has a negative aspect. All of this is consistent with what is presented in the literature related to Big Data potential and its threats [28,29].

On the other hand, the latent structure of VABIDAE is coherent with Rosenberg and Hovland's three-dimensional models of attitude [30]. According to this, the attitude is a three-dimension-based construct: cognition, affection and behaviour. VABIDAE includes questions about the participants' thoughts about Big Data (cognition), and about what and how they feel about Big Data (emotion). Also,

it includes one question about intention; however, it would be difficult to consider it as a behavioural indicator. It would be worth studying this relationship in more detail, considering that attitudes have been shown as a key factor in making the decision to use information and communication technologies in the classroom [31].

Regarding the second aim, this research also studied the profile of the participants by countries, because the differences between countries are evident [32]. Nevertheless, it can be said that a more positive than negative view predominates over the potential of Big Data for educational purposes. Also, considering that participants point out that Big Data could improve teaching, the predisposition, as in other technologies, was to be expected [33].

Although all results are very promising, it is important to take into consideration that the sample could be small and potentially unstable in replication studies. According to the results, the VABIDAE six-structure factor should be considered. Besides, new research should be developed with a more heterogeneous sample that includes students of other countries and educational disciplines. Also, it would be necessary to analyse VABIDAE including moral and ethical elements, coping values, pedagogical adjustment, etc., broadening its evaluation spectrum.

On the other hand, the VABIDAE structure seems to be flexible enough to measure the opinion, perspective and confrontation of the educators with other emerging technologies, such as virtual reality, augmented reality or educational robotics. For this purpose, versions of VABIDAE should be developed by adapting the items to that new technology or modify it, as necessary, so it may be applied independently of the technology analysed.

Therefore, VABIDAE is valid (in terms of measurement and assessment) to evaluate the future educators' stance on Big Data. In this sense, it is an instrument that can help the implementation of Big Data in schools, in a correct way. The data generated in the classroom has to be managed by the decision-makers. This includes school managers and teachers. The inclusive school benefits from the real-time knowledge of the social dynamics that take place in each specific school. Therefore, Big Data is the optimal medium for inclusive decision-making. However, the commitment of teachers and managers (teachers, managers and stakeholders in general) is indispensable. Only in this way can the challenge of inclusion in education be efficiently addressed, and with it, sociocultural sustainability.

In short, via data collected through different platforms and technological applications of administrations and schools, this medium can provide valuable information to those responsible for establishing educational policies, curriculum adaptations or educational support programs. To do all this (social sustainability from education), it is necessary to have instruments that measure the level of knowledge, opinions and emotions that educators have in relation to this technology, and therefore we have to create instruments and validate them.

## 5. Conclusions

As a conclusion, it can be stated that the results show that VABIDAE can be used as an instrument to measure how future educational professionals perceive and confront Big Data in Education. This scale could be useful in higher educational institutions, mainly related to education teaching, to help teachers and educational managers to know the standing and attitudes of their students and teachers, professors and lecturers on this technology, and then to make decisions regarding how to implement it.

Likewise, it seems that a generic instrument can be developed that values the position of the educator in relation to new technologies based on VABIDAE, which is especially important in a socioeducational reality of continuous change.

Big Data is a powerful new ally that allows new knowledge and systems to be generated—in real time—to improve decision-making. This is fundamental in teaching and learning processes.

Finally, it is important to emphasize the willingness of students to integrate this technology into their professional reality even though they are aware of its problems and risks.

**Author Contributions:** Conceptualization, J.J.L.-O. and A.M.-T.; methodology, analysis, and results A.M.-T.; writing—original draft preparation, P.D.F.-C.; writing—review and editing, J.J.L.-O. and F.J.G.-A. All authors have read and agreed to the published version of the manuscript.

**Funding:** This research did not receive any external funding.

**Conflicts of Interest:** The authors declare no conflict of interest.

## Appendix A

Parameter Estimates (with Robust Standard Errors)

Estimate Corrected SE z value Pr(> |z|)

lam[v25:Negative.Feeling]  0.891952529  0.04695508  18.9958674  1.845135e−80  v25  <— Negative.Feeling

lam[v26:Negative.Feeling]  0.855904706  0.05391615  15.8747375  9.480665e−57  v26  <— Negative.Feeling

lam[v27:Negative.Feeling]  0.965003937  0.04316316  22.3571184  1.029488e−110  v27  <— Negative.Feeling

lam[v28:Negative.Feeling]  0.857840500  0.05075187  16.9026378  4.302223e−64  v28  <— Negative.Feeling

lam[v30:Negative.Feeling]  0.916823293  0.04829774  18.9827366  2.369280e−80  v30  <— Negative.Feeling

lam[v31:Negative.Feeling]  0.684276895  0.05526303  12.3821813  3.263508e−35  v31  <— Negative.Feeling

lam[v14:Negative.Impacts]  0.679420205  0.05353211  12.6918245  6.563866e−37  v14  <— Negative.Impacts

lam[v15:Negative.Impacts]  0.657073045  0.05451117  12.0539151  1.849517e−33  v15  <— Negative.Impacts

lam[v16:Negative.Impacts]  0.699263248  0.05415056  12.9133155  3.786395e−38  v16  <— Negative.Impacts

lam[v17:Negative.Impacts]  0.626034393  0.04798024  13.0477558  6.544696e−39  v17  <— Negative.Impacts

lam[v18:Negative.Impacts]  0.822262169  0.04575593  17.9706130  3.310364e−72  v18  <— Negative.Impacts

lam[v19:Negative.Impacts]  0.881684001  0.04633771  19.0273540  1.012370e−80  v19  <— Negative.Impacts

lam[v20:Negative.Impacts]  0.882741273  0.04799176  18.3936023  1.478237e−75  v20  <— Negative.Impacts

lam[v21:Negative.Impacts]  0.863338310  0.04685169  18.4270471  7.971722e−76  v21  <— Negative.Impacts

lam[v1:Positive.impacts] 0.663363167 0.05648797 11.7434405 7.631731e−32 v1 <— Positive.impacts

lam[v2:Positive.impacts] 0.690942835 0.05597352 12.3441016 5.241608e−35 v2 <— Positive.impacts

lam[v3:Positive.impacts] 0.718533590 0.05827112 12.3308704 6.177583e−35 v3 <— Positive.impacts

lam[v8:Positive.impacts] 0.634469034 0.05019830 12.6392529 1.282670e−36 v8 <— Positive.impacts

lam[v10:Positive.impacts]  0.826951986  0.04549091  18.1784008  7.653329e−74  v10  <— Positive.impacts

lam[v11:Positive.impacts]  0.810659822  0.04987037  16.2553406  2.047273e−59  v11  <— Positive.impacts

lam[v4:Educational.Improving]  0.664861387  0.04667780  14.2436303  4.910825e−46  v4  <— Educational.Improving

lam[v5:Educational.Improving]  0.632280262  0.05554509  11.3831897  5.071040e−30  v5  <— Educational.Improving

lam[v6:Educational.Improving] 0.769933028 0.05117465 15.0452032 3.711921e−51 v6 <— Educational.Improving

lam[v7:Educational.Improving] 0.890776666 0.04485963 19.8569783 9.591023e−88 v7 <— Educational.Improving

lam[v9:Educational.Improving] 0.674736787 0.05262541 12.8215015 1.242705e−37 v9 <— Educational.Improving

lam[v22:Positive.Feelings] 0.710587212 0.05579228 12.7362996 3.716062e−37 v22 <— Positive.Feelings

lam[v23:Positive.Feelings] 0.982290598 0.04061165 24.1874069 3.018700e−129 v23 <— Positive.Feelings

lam[v24:Positive.Feelings] 0.933291957 0.04266089 21.8769904 4.303205e−106 v24 <— Positive.Feelings

lam[v29:Positive.Feelings] 0.759183236 0.04603212 16.4924690 4.156135e−61 v29 <— Positive.Feelings

lam[v12:Privacy] 1.091805603 0.05269855 20.7179454 2.386539e−95 v12 <— Privacy

lam[v13:Privacy] 0.878050897 0.05197402 16.8940339 4.978006e−64 v13 <— Privacy

C[Negative.Feeling,Negative.Impacts] 0.289448668 0.05896168 4.9090984 9.149606e−07 Negative.Impacts <–> Negative.Feeling

C[Negative.Feeling,Positive.impacts] −0.123581828 0.05617554 −2.1999225 2.781239e−02 Positive.impacts <–> Negative.Feeling

C[Negative.Feeling,Educational.Improving] −0.008988122 0.05945366 −0.1511786 8.798348e−01 Educational.Improving <–> Negative.Feeling

C[Negative.Feeling,Positive.Feelings] −0.170786261 0.06120965 −2.7901853 5.267788e−03 Positive.Feelings <–> Negative.Feeling

C[Negative.Feeling,Privacy] 0.164444579 0.05709107 2.8803908 3.971825e−03 Privacy <–> Negative.Feeling

C[Negative.Impacts,Positive.impacts] 0.091595300 0.07087262 1.2923933 1.962210e−01 Positive.impacts <–> Negative.Impacts

C[Negative.Impacts,Educational.Improving] 0.122113163 0.07205939 1.6946183 9.014785e−02 Educational.Improving <–> Negative.Impacts

C[Negative.Impacts,Positive.Feelings] −0.152945875 0.06518782 −2.3462341 1.896419e−02 Positive.Feelings <–> Negative.Impacts

C[Negative.Impacts,Privacy] 0.633223641 0.04178657 15.1537606 7.155244e−52 Privacy <–> Negative.Impacts

C[Positive.impacts,Educational.Improving] 0.837229219 0.02738604 30.5713895 2.939071e−205 Educational.Improving <–> Positive.impacts

C[Positive.impacts,Positive.Feelings] 0.663884124 0.03568547 18.6037661 2.995081e−77 Positive.Feelings <–> Positive.impacts

C[Positive.impacts,Privacy] 0.014851235 0.06419725 0.2313376 8.170526e−01 Privacy <–> Positive.impacts

C[Educational.Improving,Positive.Feelings] 0.647193563 0.03790460 17.0742763 2.306681e−65 Positive.Feelings <–> Educational.Improving

C[Educational.Improving,Privacy] −0.016140797 0.06666895 −0.2421037 8.086998e−01 Privacy <–> Educational.Improving

C[Positive.Feelings,Privacy] −0.214979070 0.05843497 −3.6789457 2.342001e−04 Privacy <–> Positive.Feelings

V[v25] 0.569193673 0.04068627 13.9898213 1.798632e−44 v25 <–> v25

V[v26] 0.802311430 0.08041357 9.9773145 1.915819e−23 v26 <–> v26

V[v27] 0.285031496 0.04978770 5.7249380 1.034715e−08 v27 <–> v27

V[v28] 0.368514865 0.03509885 10.4993433 8.698313e−26 v28 <–> v28

V[v30] 0.769775623 0.08196599 9.3914023 5.920666e−21 v30 <−> v30
V[v31] 0.862342863 0.07189301 11.9948073 3.782963e−33 v31 <−> v31
V[v14] 0.892916707 0.05996888 14.8896676 3.846775e−50 v14 <−> v14
V[v15] 0.776535610 0.06320344 12.2862875 1.073226e−34 v15 <−> v15
V[v16] 0.969240849 0.05602844 17.2990855 4.779152e−67 v16 <−> v16
V[v17] 0.544548052 0.03451537 15.7769719 4.481796e−56 v17 <−> v17
V[v18] 0.515633160 0.04966989 10.3812029 3.019463e−25 v18 <−> v18
V[v19] 0.458892061 0.04415766 10.3921293 2.692735e−25 v19 <−> v19
V[v20] 0.425799031 0.05027194 8.4699137 2.455750e−17 v20 <−> v20
V[v21] 0.469886808 0.05090533 9.2306016 2.691162e−20 v21 <−> v21
V[v1] 0.287517445 0.02303710 12.4806268 9.523404e−36 v1 <−> v1
V[v2] 0.376879371 0.04549877 8.2832878 1.198204e−16 v2 <−> v2
V[v3] 0.382960488 0.03633112 10.5408395 5.599681e−26 v3 <−> v3
V[v8] 0.385759477 0.04118148 9.3673050 7.440816e−21 v8 <−> v8
V[v10] 0.268496114 0.02904958 9.2426838 2.403906e−20 v10 <−> v10
V[v11] 0.535726031 0.06282283 8.5275692 1.494549e−17 v11 <−> v11
V[v4] 0.536110228 0.04667972 11.4848633 1.571839e−30 v4 <−> v4
V[v5] 0.738009692 0.05727916 12.8844354 5.507711e−38 v5 <−> v5
V[v6] 0.398159536 0.03943693 10.0961082 5.747816e−24 v6 <−> v6
V[v7] 0.500320312 0.05109639 9.7916965 1.222276e−22 v7 <−> v7
V[v9] 0.831292681 0.06803494 12.2186141 2.472640e−34 v9 <−> v9
V[v22] 0.883788831 0.09958798 8.8744527 7.027812e−19 v22 <−> v22
V[v23] 0.246828514 0.03701889 6.6676371 2.599546e−11 v23 <−> v23
V[v24] 0.469080598 0.05133893 9.1369375 6.424590e−20 v24 <−> v24
V[v29] 0.843168386 0.06633800 12.7101865 5.191050e−37 v29 <−> v29
V[v12] 0.212970914 0.09192344 2.3168293 2.051303e−02 v12 <−> v12
V[v13] 0.560134415 0.07336131 7.6352834 2.253243e−14 v13 <−> v13

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
