# Peer review of "Validity of the “Big Data Tendency in Education” Scale as a Tool Helping to Reach Inclusive Social Development"

_sustainability, doi:10.3390/su12135470_

Round 1

Reviewer 1 Report

As I wrote previously, the statistical analysis is provided appropriately and supports the conclusion of the paper. It seems to me that such a methodology could be interesting.

Author Response

First of all, we want to thank you for your comments., and we are in the hope the paper will be published finally.

Yours sincerely.

Reviewer 2 Report

Not clear how VABIDAE questions relate to the topic stated in the new title : “helping to reach the Inclusive Social Development”.

The section methodology is not clear.

The authors are recommended to take into consideration published research results: https://www.researchgate.net/publication/340655180_Big_Data_in_Education_Perception_of_Training_Advisors_on_Its_Use_in_the_Educational_System

Big Data in Education: Perception of Training Advisors on Its Use in the Educational System

April 2020  DOI: 10.3390/socsci9040053

Author Response

First of all, we want to thank you for your comments. Addressing these concerns in a new version of the manuscript has allowed us to improve its quality.

In the following paragraphs, we highlight the changes included in this new version of our manuscript and how we have addressed the comments. In the following you will find comments in italics followed by our response in current font.

It is a sincerely pleasure to give answers, feedback, comments and debate all those interesting suggestions. Let us go one by one.

-----------------

Reviewer 2. Lines 2224: As a result, the profile defining the participants in relation to Big Data was identified. Finally, the implications of the Big Data for Inclusive Education in a sustainable society are discussed. Can the authors specify their research objectives?

- You can find in line 16 and 76. Because of in the abstract it could be not clear, we have type the word “aim is…” to clarify.

Reviewer 2. What section in Introduction or another relates to lines 2224?

- The issue is developed in Lines 254 and followings.

Reviewer 2. Would authors like to specify what is meant by Inclusive Education in relation to their research?

The reviewer comment note we that perhaps, the Inclusive School is not a concept as general as we thought, so, we have included some phrases in order to clarify in lines 46 and following.

Reviewer 2. How it affects the research objectives?

The relation is inverse. Because of the Big Data (BD) is affecting to Education in general, it is affecting to Inclusive Education too, and the affection will be positive, but depending on how educators deal with BD. So, having instruments as VABIDAE is important to know the state of the future educational students regarding the BD. This idea is developed in line 50 and following. Nevertheless, line 55 has been enriched.

Reviewer 2. Was the authors` intention to investigate what educators think in case if educators` judgements are based on watching the video or practical experience? It feels the context of educators experience and academic environment needs elaboration. This elaboration might help for further discussion of methodology (first paragraph may carry more specific information in relation to this research) and results.

The VABIDAE authors explain on the scale website that the video is included because usually, people do not know what the BD is. So, the video focuses the concept on. The aim of this study is to validate the Scale with this sample as the first step to developing a course. This is developed on line 62 and followings.

The scale, as a unit (included video), is validated, and on line 244 is debated its potential to.

Reviewer 2. As Page 5 Table 1 the statements: I feel guilty; I feel shame, etc. do not immediately relate to any possible research objectives (they are not mentioned in the paper. In Introduction many options of possible objectives are mentioned).

I appreciate this comment because it let us bring out the concept of symmetry in Likert Scales. Of course, it is not related to any specific goal of this study, but to the whole VABIDAE validation with this specific sample. There is no doubt there is an abuse of the Likert Scale and how they are (bad) made (e.g. even number of options instead of an odd number, or no respecting the symmetry assumption, among others). VABIDAE follows the Likert criteria, mainly the symmetry properties, positive and negative sentences, etc. In this case, because these topics are in many books about methodology, and because they are not linked with the aim of this study, we would like not to have to include this debate in the paper.

Reviewer 2.Lines 28-33 seem to need further elaboration in theory support and grounded judgement.

The paragraph has been removed because of its role was only introducing the issue.

Reviewer 2. How is the following related to possible research objectives: lines 4750: the benefits of Big Data technology in Education would be linked to the improvement of education by helping personalized training, guiding students towards the programs that best fit their needs, effectively linking students to the labour market, making educational funding more transparent, and improving the administration of the educational system. On the whole the ideas expressed in lines 4750 are quite sound but how are they related to the research findings? If case the research is focused on educators using the big data tools, is it possible to make assumptions how it benefits the learners in logic with the lines 4750?

The paragraph tries to give a supporting, a context, in order to show that BD is a cornerstone in the future of inclusive education. We expect, the modification incorporated because of the previous comment turn it into clearer.

Reviewer 2. The above seems to be inconsistent with the lines 5557: Considering this situation, to achieve social sustainability from Education it is necessary to have instruments that measure the level of knowledge, opinions and emotions that educators have in relation to this technology, to be sure that it will be used correctly. Have the authors found the instruments? What was the intention behind lines 55-57? Further elaboration may be needed.

This sentence advances the idea of the both next paragraphs, introducing the part about justifying the aim (that is in the last paragraph). In this sense, I appreciate all you keep in mind that we follow the basic scheme provided by APA institution. In order to attend the comment, it has been introduced a brief but significative sentence based on the references number 2, 4 and 5.

Reviewer 2. Lines 5154: Please provide references supporting the opinions, e.g. line 51 most vulnerablewhat category of students and why is it taken into consideration, how it relates to the research discussed.

The reference is the 8th. It has been move on. On the other hand, text is speaking about School in general (line 47). That is because of it is not specified any “category” of students.

Reviewer 2. Line 52: increase the digital gap. Again, these are good pieces of text but they need specification to the research context. Please check for such pieces of texting throughout the paper.

Again, we appreciate this suggestion that increases the quality of the text. It has been reviewed and removed where it did not make sense.

Reviewer 2. In this respect, it is not clear how VABIDAE questions relate to the topic stated in the new title: helping to reach the Inclusive Social Development”. and to line 70: In relation to Big Data in Education focusing on educators, students and managers.

In general, the idea behind the text is: BD will be important in Education, its impact will be positive if educator manage it properly, it will positive if educators receive a well supported training, the first step is to have assessment instrument. We hope, the new sentences and modifications improve the understanding.

Reviewer 2. Lines 9293 may help to clarify the Introduction: Big Data Applied to Education Scale (VABIDAE) [13] is a 31 items questionnaire that gathers information about opinions and how people face and perceive the presence of Big Data technologies in the Educational System and in classrooms.
If it is the intention, it may be helpful to make it clear why educators answer the questionnaire and how it is presumed to help the students. From here, certain aspects of Big data for education may become more specific in the Introduction.

Please specify if it is pilot study. This might help to bring consistency into research description.
In section Results it is also not clear in relation to what research objectives the results are obtained. Although the discussion of results has been put in fine writing.

It has been included that it is a pilot study. We hope that the modifications, because of previous suggestions, are given more coherence.

Reviewer 2. Section Discussion. Please specify, as it is not clear from the sections introduction and research methods: in case educators have not have previous experience in applying Big Data, what coherence of results is meant in lines 213215: The results show that VABIDAE has high internal consistency, showing coherence about the Big Data effects, both positive (teaching improvement, curricular adaptation, etc.) and negative (privacy issues, isolation, etc.). It seems further elaboration and clarification is needed.

In this case, the internal consistency construct is debated related to accuracy measurement. That is because “both positive… negative..” effects are brought up. To avoid misunderstanding, we have added the word “measurement”.

Reviewer 2. Lines 233234: This research has also studied the profile of the participants by countries, because the differences between countries are evident.
Do authors intend to elaborate on this statement in relation to what research objectives?

Thank you for the note. It is in relation to second aim (line 78). It has been added “about the second aim…”.

Reviewer 2. What is meant by the authors? lines 249250: Therefore, VABIDAE is an instrument that can help the implementation of Big Data in schools, in a correct way.

We want to say in short, that Vabidae is valid (in terms of measurement and assessment) to evaluate the future educators stance on BD. We have added this sentences.

Reviewer 2. On the whole research description needs clear objectives and it would be beneficial to dwell more on academic settings specifics; interrelating the methodology, results and research objectives.

We are in hope all modifications incorporated have satisfaced the weaknesses and have improved the quality of text.

Reviewer 2. As for application of VABIDAE, the authors are recommended to take into consideration published research results: https://www.researchgate.net/publication/340655180_Big_Data_in_Education_Perception_of_Traini ng_Advisors_on_Its_Use_in_the_Educational_System

Big Data in Education: Perception of Training Advisors on Its Use in the Educational System April 2020 DOI: 10.3390/socsci9040053

We are really pleased for this comment. Nevertheless, because of conflict of interest could be implicated, and taking into the editor was notified, we would like not say anything about this comment.

Yours sincerely.

Round 2

Reviewer 2 Report

The overall quality of the text does not seem to be improved since the previous reviews, based on the authors responses.

Author Response

A new revised version of our manuscript entitled “Validity of the ‘Big Data Tendency in Education’ Scale as a tool helping to reach the Inclusive Social Development” has been sent.

We appreciate the overall your suggestions of the manuscript and agree with the concerns raised by you. Addressing these concerns in a new version of the manuscript has allowed us to improve its quality.

We are sure that if you are so kind of consult the track record in MSWord, you will appreciate the improvement of the content and the usefulness of the result in order to increase the corpus of knowledge in Education Sciences into our small field of specialization.

Sincerely, yours
